# Contribution of women's preference to the overuse of caesarean sections: A propensity score matching analysis based on a multi-country cross-sectional survey, as part of the QUALI-DEC project

Camille Etcheverry[ID][1]*, Ana Pilar Betrán[2], Marion Ravit[1], Charles Kaboré[3�],
Pisake Lumbiganon[4�], Guillermo Carroli[5�], Quoc Nhu Hung Mac[6�], Celina Gialdini[5,7�],
Alexandre Dumont[1], the QUALI-DEC research group[¶]

1 Université Paris Cité and Université Sorbonne Paris Nord, IRD, Inserm, Ceped, F-75006 Paris,
France, 2 UNDP/UNFPA/UNICEF/WHO/World Bank Special Programme of Research, Development and
Research Training in Human Reproduction (HRP), Department of Sexual, Reproductive, Maternal, Child,
Adolescent Health and Ageing, World Health Organization, Geneva, Switzerland, 3 Institut de Recherche
en Sciences de la Santé, Ouagadougou, Burkina Faso, 4 Department of Obstetrics and Gynaecology,
Faculty of Medicine, Khon Kaen University, Khon Kaen, Thailand, 5 Centro Rosarino de Estudios
Perinatales, Rosario, Argentina, 6 Pham Ngoc Thach University, Ho Chi Minh City, Viet Nam, 7 Facultat
de Ciències de la Salut Blanquerna, Universitat Ramon Llull, Barcelona, Spain

¶ Membership of the QUALI-DEC consortium group is provided in the Acknowledgments.
☯ These authors contributed equally to this work.
* camille.etcheverry@ird.fr

## Abstract

### Introduction

Maternal request for caesarean section has often been cited to justify the increasing caesarean section rates worldwide. However, we lack evidence on the impact of women's preference for caesarean section on this dramatic tendency. Given the need to develop appropriate strategies to reduce unnecessary caesarean section, the objective of this study was to assess the association between women's preference for caesarean section and its actual use, and to estimate the proportion of caesarean section associated with women's preference for caesarean section in Argentina, Burkina Faso, Thailand and Viet Nam.

### Methods

A cross-sectional hospital-based survey among postpartum women was conducted in 32 hospitals (8 per country) between 2020 and 2022. We selected women with no potential medical indication for caesarean section among a random sample of women who delivered in participating facilities during the data collection period. We chose a propensity score matching approach, to compare the probability of giving birth by

**Data availability statement:** The data analyzed in this study, derived from four cross-sectional surveys conducted as part of the QUALI-DEC project, are available on Zenodo via the following web links: https://doi.org/10.5281/zenodo.17386804 for Argentina; https://doi.org/10.5281/zenodo.17386857 for Burkina Faso; https://doi.org/10.5281/zenodo.17386869 for Thailand; https://doi.org/10.5281/zenodo.17386880 for Viet Nam.

**Funding:** The QUALI-DEC project is co-funded by the European Union's Horizon 2020 research and innovation program under grant agreement No. 847567 and by the UNDP-UNFPA-UNICEF-WHO-World Bank Special Programme of Research, Development and Research Training in Human Reproduction (HRP), a cosponsored program executed by the World Health Organization (WHO) in the Department of Sexual and Reproductive Health and Research (SRH). The contents of this article are solely the responsibility of the authors and do not reflect the views of the EU, UNDP, UNFPA, UNICEF, WHO, or the World Bank or their respective institutions. The first author (C.E.) received salary support from the Ecole Doctorale Pierre Louis de Santé Publique (Sorbonne Université Université Paris Cité) as part of a thesis funding. The funders had no role in study design, data collection and analysis, decision to publish, or preparation of the manuscript.

caesarean section between women who, late in pregnancy, preferred caesarean section and those who preferred vaginal birth.

## Results

A total of 1,827 low-risk women were included, of whom 10.4% preferred a caesarean section and the average caesarean section rate was 24.5%. The results show that, on average, preference for caesarean section increased the probability of having a caesarean section by 32% (CI 95% [0.23–0.41]; p < 0.001). The relative risk was estimated at 2.69 (CI 95%: 2.43; 2.95) and the fraction of caesarean section associated with women's preference was estimated at 15% (CI 95%: 12.9% − 16.9%).

## Conclusion

Although women's preference plays a role in the use of caesarean section in the participating hospitals, it likely accounts for only a small proportion of the caesarean section performed, highlighting the need for multidimensional, context-specific strategies to reduce unnecessary caesarean sections (providing women with evidence-based information, improving clinicians' adherence to guidelines and shared decision-making, addressing systemic factors…).

---

## Introduction

Over the past 30 years, there has been a significant increase in the use of caesarean section (CS) in many parts of the world [1–3]. From an global rate of 6.7% in 1990 to 21.1% in 2018, some regions of the world have seen a spectacular increase in their CS rates, such as East Asia (from 4.9% in 1990 to 33.7% in 2018) and Latin America (from 22.8% in 1990 to 42.8% in 2018) [1,2]. This trend affects many low- and middle-income countries (LMICs), where it is common to observe both an abusive practice of CS among women at low obstetrical risk and predominantly socioeconomically advantaged, and an under-use of this procedure among the most disadvantaged, due to a lack of access to care [4,5]. However, there is no benefit in performing a CS in the absence of a medical indication, and particularly in resource-limited settings where it may increase morbidity and mortality [6,7]. The increase in CS represents a serious challenge for LMICs because of the increase in maternal and perinatal morbidity associated with its excessive use, the health inequalities it causes and the resulting diversion of otherwise limited resources [4,8–10].

There are many reasons for the excessive use of CS [8]. As summarized by the ecological model of Betrán et al., many non-clinical factors related to clinicians' practices (e.g., loss of skills in conducting vaginal deliveries, clinicians' preference for CS) and to health system (e.g., financial incentives, lack of institutional guidelines) influence the use of CS [8]. Besides, mothers' demand for this mode of birth has often been cited to justify the overall increase in CS rates [8,11]. Indeed, this demand seems to be associated with non-medically justified use of CS in certain contexts

[12]. Women's demand for CS is often motivated by fear of pain and complications associated with vaginal birth, as well as by the perceived advantages of CS, such as the possibility of planning the birth and the supposed safety of this mode of delivery [8,13–19]. According to Panda et al., maternal request for CS may then encourage clinicians to perform the procedure, mainly due to the fear of litigation from the women and their families or personal convictions regarding women's rights and their autonomy to choose their mode of delivery [11].

Since the number of CS performed on maternal request is often difficult to measure—particularly when this practice is not officially authorized and documented in medical records—previous studies have explored the relationship between women's preference for CS and their actual mode of birth [20–24]. Considering the preference for CS as an indicator of maternal demand, these studies aimed to assess its influence on the decision for mode of birth [20–23]. In Norway, nulliparous women who preferred cesarean delivery were significantly more likely to undergo antepartum CS, but also, to a lesser extent, intrapartum CS [23]. In China, Deng et al. estimated that the probability of giving birth by CS on maternal request increased when the woman had a preference for CS in late pregnancy [22]. However, this link may vary depending on the context, and on whether women's preference is taken into account in the decision on the mode of delivery [25,26].

In addition, several studies have concluded that women's preference alone cannot explain the dramatic increase in CS rates, as the proportion of women expressing this preference remains low [13,25–27]. In 2011, a meta-analysis revealed that approximately 22% of women in LMICs preferred CS [27]. More recently, a systematic scoping review indicated that the proportion of women preferring CS varied from 1.4% to 50% in these countries [13], highlighting the heterogeneity of contexts and the different methodologies used to measure women's preferences for this mode of delivery [13]. Thus, questions remain regarding the actual impact of women's preference and demand for CS on the increasing overuse of CS observed in many LMICs.

**The QUALI-DEC project**

In response to the significant increase in CS rates worldwide, interventions aimed at reducing unnecessary CS target pregnant women, healthcare providers, and healthcare systems [8,28]. For women, these mainly consist of educational activities, childbirth preparation workshops, and ongoing support during labor [29,30]. For professionals, the most effective strategies include the implementation of evidence-based guidelines and clinical audits [31], while financial or regulatory measures are not supported by strong evidence [32].

The most promising strategies must combine several interventions involving all stakeholders, addressing non-clinical determinants and adapted to local contexts in LMICs. That is why a consortium of researchers has developed the QUALI-DEC project (Appropriate use of CS through QUALIty DECision-making by women and providers), aimed at implementing and evaluating evidence-based non-clinical interventions to reduce the number of CS performed on low-risk women in four LMICs [33]. Women belonging to groups 1–4 of the Robson classification are the target population for Quali-Dec [34]. They are at lower risk of CS, as compared to groups 5, or groups 6–10. The project was carried out in 32 facilities with high CS rates in Argentina, Burkina Faso, Thailand and Viet Nam. The QUALI-DEC project included four non-clinical interventions: (1) opinion leaders to implement evidence-based clinical guidelines; (2) CS audits with feedback to help providers identify potentially avoidable CS; (3) implementation of WHO recommendations on labour companionship to support women during vaginal birth; and (4) a decision analysis tool to help women make informed decisions about mode of delivery [33]. Used during prenatal care visits, the Decision Analysis Tool (DAT) is both a way to inform low-risk women about the risks and benefits of each mode of delivery and a tool to initiate and support dialogue between these women and their healthcare providers, encouraging shared decision-making about the delivery mode [35].

Previous analysis showed that women's preference for CS was low in the hospitals participating in the QUALI-DEC project, although it varied between countries [36]. Having a preference for CS was more common among nulliparous women and was linked to women's fear of pain and childbirth and to doctors' influence in promoting CS, especially in Viet Nam [36]. While this preference alone does not appear to fully explain the high CS rates observed in the participating

hospitals, we aimed to explore whether, and to what extent, it plays a determining role in the decision-making process regarding the mode of delivery. Additionally, the research team sought to assess how the Decision Analysis Tool could help reduce non-medically justified CS, based on the assumption that a better understanding of the risks and benefits of each delivery method – facilitated by this tool – might encourage some women who preferred CS to reconsider their choice.

This study aimed to improve understanding of the role of women's preference for CS in its overuse, to guide strategies to reduce unnecessary CS. To this end, the objective of this study was to assess the association between women's preference for CS and its actual use, and to estimate the proportion of CS associated with women's preference for CS in Argentina, Burkina Faso, Thailand and Viet Nam.

## Materials & methods

### Ethical considerations

Scientific and ethical approvals were obtained from the following institutions, in accordance with the Declaration of Helsinki: 1) Ethics Committee for Health Research of Burkina Faso (Decision No. 2020-3-038), 2) the Research Project Review Panel (RP2) in the UNDP/UNFPA/UNICEF/WHO/World Bank Special Programme of Research, Development and Research Training in Human Reproduction (WHO Study A66006) at the WHO, 3) the WHO Research Ethics Review Committee (ERC), Geneva, Switzerland, 4) the French Research Institute for Sustainable Development, 5) the Central Research Ethics Committee; CREC (Certificate Number COACREC002/ 2021) in Thailand, 6) Department of Reproductive Health of the Ministry of Health in Viet Nam, and 7) Centro Rosarino de Estudios Perinatales of Rosario, Argentina (Record Notice No. 1/20). All procedures were conducted in accordance with relevant guidelines and regulations. To ensure data anonymization, each participant was assigned a unique study identification number. Written informed consent was obtained from all participants prior to survey, which was carried out in a private setting within the hospital to ensure the confidentiality of the interviews. Additional information regarding the ethical, cultural, and scientific considerations specific to inclusivity in global research is included in the Supporting Information (S1 Checklist).

### Study design

This paper is derived from an ancillary analysis to the QUALI-DEC project, which is a type III hybrid efficacy-multi-site trial conducted in Argentina, Thailand, Viet Nam and Burkina Faso and registered on the Current Controlled Trials website (ISRCTN67214403) [33,37]. The primary objective of this trial is to evaluate the effect of the QUALI-DEC strategy on CS rates and maternal and perinatal outcomes. The detailed methodology of the trial has been published elsewhere [33,37]. The project design includes two cross-sectional surveys among a representative sample of postpartum women (before and after the intervention period), allowing a before-and-after comparison. This ancillary study used data from the baseline cross-sectional survey (before the interventions). These data were collected during the baseline period in 32 hospitals in Argentina, Burkina Faso, Thailand and Viet Nam (8 per country). The participating hospitals were purposively selected by the health ministries of the participating countries because of their high CS rates [36] and their representativeness in terms of level of care (secondary versus tertiary referral health facilities) and mode of practice (public versus private). Average CS rates and characteristics of participating hospitals by country are presented in S2 Table (S2 Table).

### Participants and sample size

The baseline survey included postpartum women who had just given birth to a live infant beyond 22 weeks gestation (28 weeks in Burkina Faso) and who agreed to take part. For ethical reason, women were not eligible if they experienced a major health problem during labor or childbirth, gave birth to a stillborn child, or had a newborn who either died before discharge or was born with a malformation. Women who gave birth at home or in another health facility (postnatal transfer) were excluded from the survey. The sample size for the survey was based on the expected difference in satisfaction

scores between the pre- and post-intervention periods [33]. The required number of women per country was 470. Assuming a non-response rate of 10% and ineligibility of 10% of women, our target was to approach 564 women in each country (71 women per hospital).

For this analysis, we selected women from the cross-sectional survey who were at low risk of CS: women with single pregnancy, at term (37 weeks or more), cephalic presentation and no history of CS (groups 1–4 of Robson's classification). We excluded women who had no preference for a mode of birth in late pregnancy or who had undergone an emergency CS before labour, as we considered that this type of surgery was generally performed on women who had presented with a major complication (eclampsia, retroplacental haematoma, etc.), regardless of the women's preference.

## Survey process and data collection

The survey was conducted over a minimum of two-week period in each hospital, with all surveys across the 32 hospitals carried out between December 2020 and June 2022. The recruitment period took place from 8 December to 26 December 2020 in Burkina Faso, from 6 March 2021–3 January 2022 in Thailand, from 15 December 2021 to 23 June 2022 in Argentina and from 8 October to 21 October 2021 in Viet Nam.

In each hospital, data collection for the baseline cross-sectional survey of postpartum women took place every day, including weekends until the required number of participants (n = 71 per hospital) was reached. If the required number of participants was reached (n = 71 per hospital) before the two-week period, data collection continued until the end of the predefined period. From the list of women who had given birth the previous day, between 5 and 6 postpartum women had to be interviewed each day to reach the required sample size. In hospitals with more than 10 deliveries per day, a randomization factor was applied each day to all women who had given birth the previous day in order to obtain a random sample of 10 women. Assuming that 4–5 women would refuse to participate or would not be eligible from this random sample, 5–6 women would be included in the survey each day, allowing the required number of subjects to be reached. The women selected were identified by a data collector who assigned them an identification number and assessed their eligibility using a screening form. If a woman was eligible, she was approached by a social scientist who invited her to take part in the study during her stay on the postnatal ward. If she agreed to take part, an informed consent form was completed in writing, and the woman was interviewed face-to-face by the social scientist using a tablet data collection form. The questionnaire was developed based on a literature review, followed by discussion and consensus with the QUALI-DEC research team. The questionnaire was tested in the four countries and adapted where necessary. The information gathered was organized into seven modules: women's characteristics, antenatal care and preference for mode of delivery; birth outcomes; women's knowledge of modes of delivery, including risks and benefits; accompaniment at birth; women's experience of childbirth and status; gender dimensions and social equity; characteristics of wealth and out-of-pocket expenses. In order to collect women's preferences in late pregnancy, this questionnaire was constructed as follows. The interviewers first asked: 'Did you have a preferred method of delivery at the end of your pregnancy' (yes/no/don't know). The women who answered 'yes' were then asked about their preference: 'What was your preferred mode of delivery at the end of your pregnancy' (vaginal birth/CS).

For all selected women, medical history and information about pregnancy, labour and delivery were extracted from medical records by a clinical data collector and entered into a paper data collection form. Data was entered twice in each country into an electronic system designed for this study with validation checks (REDCap®). Consistency checks were managed centrally by the main data manager, with regular communication with national data managers in Thailand, Burkina Faso and Viet Nam. In Argentina, ongoing consistency checks were managed by the local team.

## Analysis methods

The analysis consisted in measuring the association between women's preference for the delivery mode (exposure) and their final mode of delivery (outcome). The exposure variable, 'preferred mode of delivery' at the end of pregnancy, was collected from women as described above. The exposure variable was defined as women's response to the question: 'What

was your preferred mode of delivery at the end of your pregnancy' (vaginal birth/CS), among women who had a preference in late pregnancy. The outcome, i.e., the participants' mode of delivery (vaginal birth or CS), was collected from the medical records. The modality 'caesarean delivery' included both CS performed before and during labour.

To take account of the complexity of the confounding factors that exist in the relationship between the exposure variable and the outcome, we chose the *propensity score matching* (PSM) approach [38]. This method restores equiprobability between women who preferred CS and women who preferred vaginal birth, by matching women who are comparable regarding the confounding factors [39,40].

Among the women surveyed at low risk of CS, with no emergency on admission and who declared a preference in late pregnancy, we considered: (i) women who had a preference for CS in late pregnancy (exposed women, $E_i = 1$); (ii) women who had a preference for vaginal birth in late pregnancy (unexposed women, $E_i = 0$). For each woman, the approach was to calculate her probability of preferring CS (propensity score) as a function of socio-demographic, clinical and institutional explanatory variables [39,40].

We made several assumptions before using the PSM approach: the positivity (for each woman, there is a non-zero probability of preferring CS or vaginal birth); the consistency (a woman's potential preference as a function of her characteristics is precisely her observed preference) and the exchangeability (women must be identical on average for the characteristics likely to influence the delivery mode, with the exception of preference) [41–44]. The absence of measurement error or interference (a woman's birth mode is not affected by the preference of other women) and the correct specification of the model were also assumed [45,46].

In order to select the variables for calculating the propensity score, we first constructed an acyclic directed graph (S3 Fig), to identify the confounding factors in the relationship between women's preference and their final mode of delivery [47]. The variables making up this graph were chosen based on pre-existing knowledge in literature, on previous results [36,48] and on information available in the survey. They include socio-demographic variables (country, urban or rural residence, maternal age, level of education, mother's occupation and wealth index); pregnancy-related variables (parity, BMI and antenatal consultations in another private institution or not); delivery-related factors (induction of labour, birth weight and presence of medical or obstetric complications); and institutional factors linked to the characteristics of the hospital where the woman gave birth (reference level, type of facility, private practice, anaesthetist dedicated to the delivery room or not, capacity of care during labour assessed based on the ratio between the average number of deliveries per day and the total number of beds in the delivery room). Some variables, such as the preference of healthcare providers, were not measured in the survey and are shown as unobserved in the graph. Based on the DAG, we entered the variables potentially associated with preference for CS (exposure) in a multilevel multivariate logistic regression model to identify those associated with p-value <0.2. Similarly, we entered the variables potentially associated with CS delivery (outcome) in a multilevel multivariate logistic regression model to identify those that were associated with p-value <0.2. In line with recommendations, we selected variables that were related to exposure and outcome or outcome only (with a p-value<0.2) using a multilevel multivariate *logit* model and a bottom-up stepwise procedure [49,50].

Statistical analyses were performed using Stata/SE® 17 software. The *pscore* command was used to calculate the propensity score and to check equiprobability between the groups. We then matched women who preferred CS to those who preferred vaginal birth with the closest propensity score. We applied the nearest-neighbour matching method with replacement, in which a woman with a preference for CS could be matched several times with a woman in the comparison group. This method was preferred because it reduced matching bias by allowing each treated unit to be matched with the closest control, even if the best-matched control had already been used. This improves the quality of the matching — particularly when the pool of control units is limited or when the treated units are difficult to match — as no treated unit is forced to accept a poorer match simply because of previous matches. Furthermore, this method avoided reducing the sample size, which was already limited (by avoiding the elimination of many treated units when no control units matched). Three checks were carried out to ensure the quality of the matching [40,51]. Firstly, we calculated the pseudo-$R^2$, based

on the variables used in the propensity score, to assess the average bias between women who preferred CS and those who preferred vaginal birth before and after matching. We ensured that the pseudo-R² decreased after matching. We then calculated Rubin's B indicator (standardized absolute difference in propensity score means between exposed and unexposed women), and Rubin's R indicator (ratio of propensity score variances in the two groups). We ensured that these indicators did not exceed 25 and 2 respectively, as recommended [40,50,51].

Finally, we assessed the association between women's preference for CS and the final mode of delivery, using the *teffect* command, which measures the mean difference in risk of giving birth by CS (and its 95% confidence interval) between matched women [52]. We first estimated this association for the whole sample, gathering women from all four countries. We then stratified the analysis by country, excluding Burkina Faso because of the small number of women who had a preference for CS. To calculate the fraction of CS associated with women's preference for this mode of delivery, we used the *nlcom* command. This allows us to estimate the relative risk (RR) of having a CS among women who preferred this mode of delivery compared with those who preferred the vaginal route, based on the potential effects calculated in each comparison group. We then applied the formula for calculating the associated fraction: $FA_p = \frac{P_p(RR-1)}{P_p(RR-1)+1}$ with $P_p$ being the proportion of women in the sample who preferred CS at the end of pregnancy.

## Results

A total of 1,827 low-risk women were included in this analysis. A flowchart describes the recruitment and selection of the study sample in Fig 1. In this sample, 10.4% of women preferred CS (14.5% in Argentina, 1.8% in Burkina Faso, 19.1% in Thailand and 5.1% in Viet Nam) and the average CS rate was 24.5% (448/1827), ranging from 18.2% in Burkina Faso to 27.6% in Viet Nam (Fig 2 and Table 1). The results show that CS rates were higher among women who preferred a CS than among those who preferred a vaginal birth (Fig 2). The Chi2 test was significant in each country (Fig 2). This difference was particularly marked in Viet Nam, where 93% of women who indicated a preference for CS had a CS delivery, compared with 23% of women who indicated a preference for vaginal birth (Fig 2).

Table 1 presents the socio-demographic and clinical characteristics of the women in the sample, as well as the institutional characteristics of the hospitals in which they gave birth, according to the exposure variable (preference in late pregnancy) and the outcome (mode of delivery). The calculation of the propensity score was based on 12 variables: maternal age, wealth index, residence, maternal profession, level of education, BMI, birthweight of the newborn, presence of complications, type of antenatal care, presence of a dedicated anaesthetist in the delivery room and capacity for care during labour (ratio between the average number of deliveries per day and the total number of beds in the delivery room).

Table 2 presents the results of the evaluation of the matching quality. The distribution of variables between the two comparison groups did not differ after matching. The mean absolute bias was less than 5%, Rubin's B was less than 25 and Rubin's R was between 0.5 and 2, as recommended. Therefore, the two comparison groups were considered sufficiently balanced.

Table 3 presents the association between women's preference and mode of delivery, in the overall sample and by country, based on the matched sample. The results show that, on average, preference for CS increased the probability of having a CS by 32% (CI95% [0.23–0.41]; p < 0.001). The relative risk was estimated at 2.69 (CI95%: 2.43; 2.95) (Table 4). Given the low proportion of women preferring CS in this sample (10.4%), the fraction of CS associated with women's preference was estimated at 15% (CI95%: 12.9% − 16.9%) (Table 4).

Analysis by country showed that preference for CS increased the probability of delivery by CS by 34% (CI95% [0.17–0.51]; p < 0.001) in Argentina; by 31% (CI95% [0.19–0.43]; p < 0.001) in Thailand; and by 58% (CI95% [0.33–0.82]; p < 0.001) in Viet Nam (Table 3). The proportion of CS associated with women's preference for CS ranged from 12% in Viet Nam to 23% in Thailand (Table 4). In other words, we estimated that 12% of CS (CI95%: 10.2%; 14.4%) in the sample were performed due to a preference for CS in Viet Nam, while this proportion was 17.8% (CI95%: 10.8%; 23.9%) in Argentina and 23% (CI95%: 18.2%; 28.0%) in Thailand (Table 4).

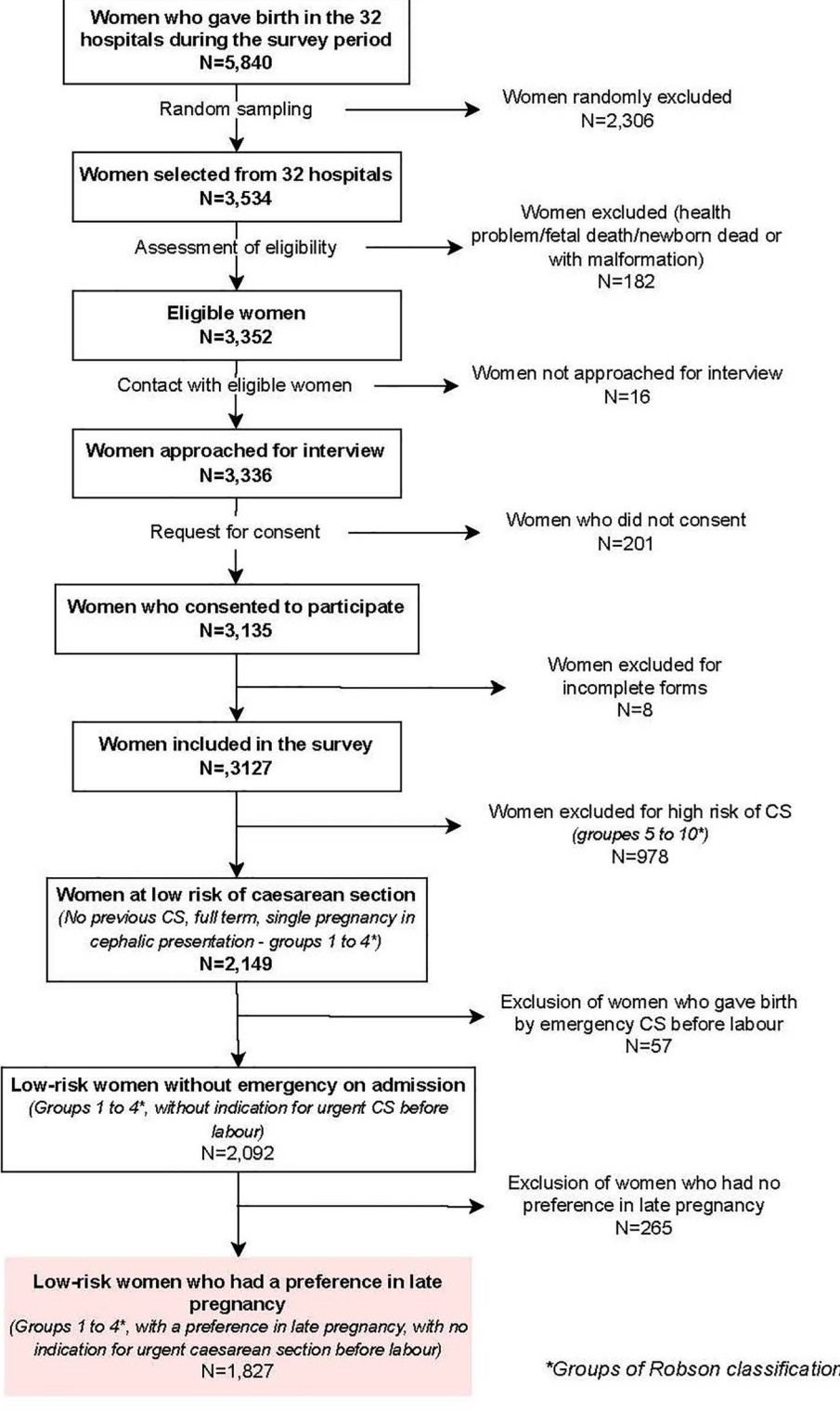

**Fig 1. Flowchart representing the recruitment and selection of included women.**

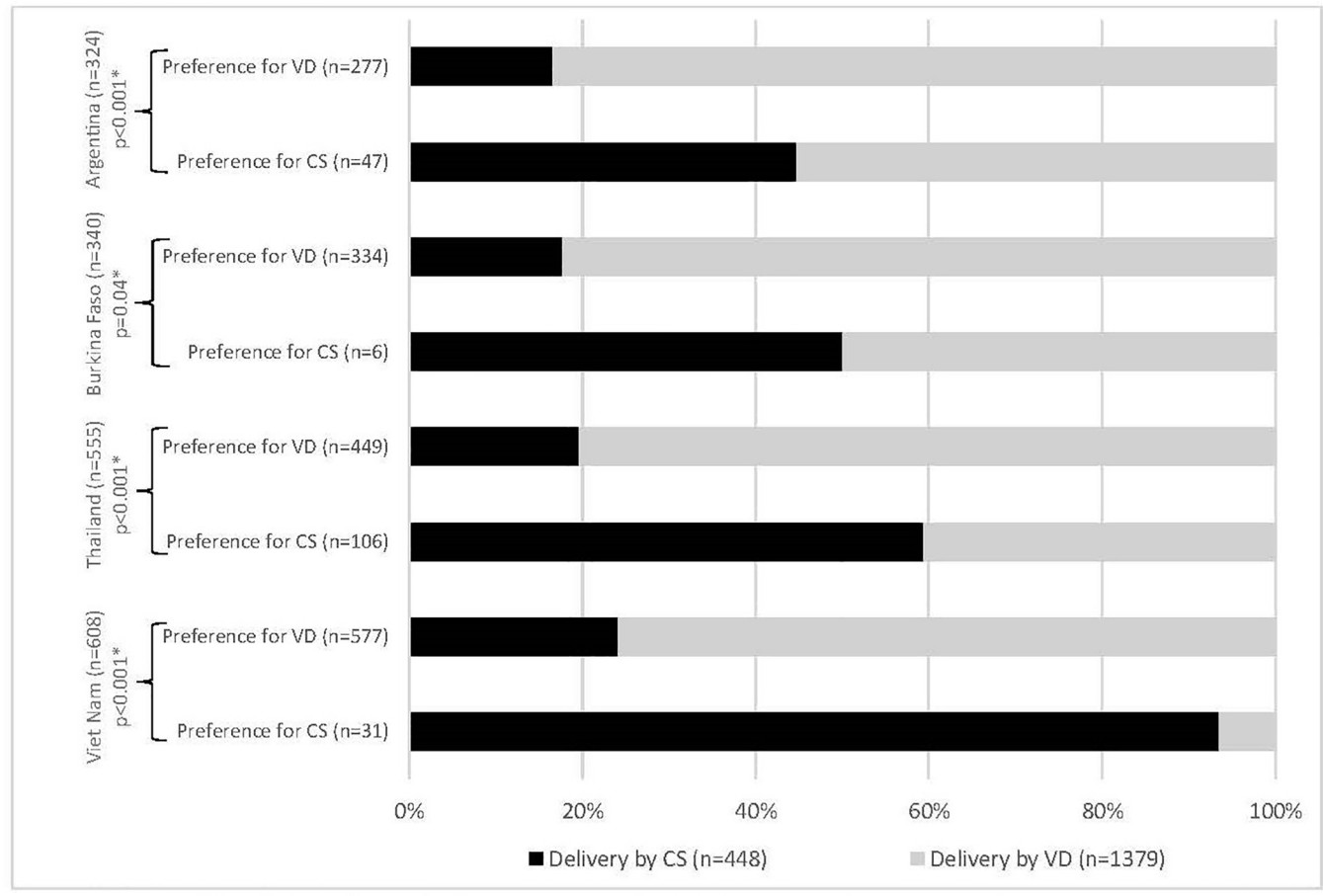

*Bivariate analyses of differences in mode of delivery according to women's preferences
CS: Caesarean section; VB: Vaginal delivery*

**Fig 2. Mode of delivery among low-risk women, according to their preference and by country (n = 1,827).**

## Discussion

In the 32 participating hospitals across Argentina, Burkina Faso, Thailand and Vietnam, the preference for CS among low-risk women appears to increase the probability of having a CS by 32% (ranging from 31% in Thailand to 58% in Vietnam). Nevertheless, the proportion of CS associated with preference for this mode of birth was calculated to be 15% (ranging from 12% in Vietnam to 23% in Thailand), which is relatively low given the low disposition of women to prefer CS and it is due to the small proportion of women preferring CS in these populations.

In LMICs, the majority of studies reporting clinicians' views describe maternal demand for CS as a 'key factor' in the observed increase in CS rates [53]. However, our findings do not confirm the perception of clinicians and suggest that women's preference for this procedure is a relatively minor contributor to the overuse of CS. Three other literature reviews, based on studies of women's demand or preference rather than clinicians' views, also suggest that women's preference for CS is only a secondary factor which, according to Gamble and Creedy, "may divert attention away from physician-led influences on the continuing high cesarean section rates" [25–27]. The high CS rates observed among low-risk women in the selected hospitals may be explained by other factors linked to the quality of care and

**Table 1. Characteristics of women included in this analysis and institutional factors according to maternal preference and mode of delivery (Quali-Dec).**

| Variables | Total (N = 1827) n (%) | Preference for CS (N = 190) n (%) | p-value* | Mode of birth by CS (N = 448) n (%) | p-value** |
|---|---|---|---|---|---|
| **Country** | | | <0.001 | | <0.01 |
| Argentina | 324 (17.7) | 47 (14.5) | | 67 (20.7) | |
| Burkina Faso | 340 (18.6) | 6 (1.8) | | 62 (18.2) | |
| Thailand | 555 (30.4) | 106 (19.1) | | 151 (27.2) | |
| Viet Nam | 608 (33.3) | 31 (5.1) | | 168 (27.6) | |
| **Maternal age** | | | 0.05 | | 0.80 |
| <25 years | 594 (32.5) | 57 (9.6) | | 140 (23.6) | |
| 25–35 years old | 966 (52.9) | 94 (9.7) | | 241 (24.9) | |
| ≥35 years | 267 (14.6) | 39 (20.5) | | 67 (25.1) | |
| **Level of education** | | | <0.001 | | <0.001 |
| Secondary and less | 1262 (69.1) | 102 (8.1) | | 262 (20.8) | |
| Superior | 565 (30.9) | 88 (15.6) | | 186 (32.9) | |
| **Place of residence** | | | <0.001 | | 0.18 |
| Rural | 507 (27.9) | 30 (5.9) | | 136 (26.8) | |
| Urban | 1311 (72.1) | 160 (12.2) | | 312 (23.8) | |
| **Mother's profession** | | | 0.85 | | 0.98 |
| Unemployed/housewife | 661 (36.2) | 70 (10.6) | | 162 (24.5) | |
| With employment | 1165 (63.8) | 120 (10.3) | | 285 (24.5) | |
| **Wealth index** | | | 0.21 | | 0.02 |
| The poorest | 401 (21.9) | 30 (7.5) | | 81 (20.2) | |
| Poor | 398 (21.8) | 43 (10.8) | | 87 (21.9) | |
| Medium | 438 (24.0) | 45 (10.3) | | 128 (29.2) | |
| Rich | 289 (15.8) | 36 (12.5) | | 74 (25.6) | |
| The richest | 301 (16.5) | 36 (12.0) | | 78 (25.9) | |
| **Parity** | | | <0.001 | | <0.001 |
| Nulliparous | 835 (45.7) | 119 (14.2) | | 295 (35.3) | |
| Multiparous | 991 (54.3) | 71 (7.2) | | 153 (15.4) | |
| **Body mass index** | | | <0.01 | | <0.001 |
| Less than 30 | 1339 (77.9) | 128 (9.6) | | 304 (22.7) | |
| 30 and over | 380 (22.1) | 58 (15.3) | | 122 (32.1) | |
| **Antenatal care in a private establishment** | | | 0.79 | | 0.001 |
| No | 1045 (57.2) | 107 (10.2) | | 225 (21.5) | |
| Yes | 782 (42.8) | 83 (10.6) | | 223 (28.5) | |
| **Presence of a complication§** | | | <0.01 | | <0.001 |
| No | 1404 (76.8) | 130 (9.3) | | 297 (21.1) | |
| Yes | 423 (23.2) | 60 (14.2) | | 151 (35.7) | |
| **Induction of labour (among women in labour, n = 1730)** | | | 0.77 | | <0.01 |
| No | 1533 (88.6) | 118 (7.7) | | 297 (19.4) | |
| Yes | 197 (11.4) | 14 (7.1) | | 54 (27.4) | |
| **Birth weight of newborn** | | | <0.001 | | <0.001 |
| Light (<2500g) | 71 (3.9) | 12 (16.9) | | 16 (22.5) | |
| Normal (2500-4000g) | 1674 (91.6) | 158 (9.4) | | 395 (23.6) | |
| Macrosomia (≥4000g) | 82 (4.5) | 20 (24.4) | | 37 (45.1) | |

*(Continued)*

**Table 1.** (Continued)

| Variables | Total (N = 1827) n (%) | Preference for CS (N = 190) n (%) | p-value* | Mode of birth by CS (N = 448) n (%) | p-value** |
|---|---|---|---|---|---|
| **Level of care** | | | <0.001 | | 0.22 |
| Primary – Secondary | 971 (53.1) | 57 (5.9) | | 227 (23.4) | |
| Tertiary | 856 (46.9) | 133 (15.5) | | 221 (25.8) | |
| **Private practice in the hospital** | | | <0.001 | | <0.001 |
| No | 845 (46.2) | 54 (6.7) | | 165 (19.5) | |
| Yes (private practice or private hospital) | 982 (53.8) | 136 (13.9) | | 283 (28.8) | |
| **Number of births per delivery bed per day** | | | 0.19 | | 0.001 |
| ≤ 2 births per bed/24h | 812 (44.4) | 76 (9.4) | | 168 (20.7) | |
| > 2 births per bed/24h | 1015 (55.6) | 114 (11.2) | | 280 (27.6) | |
| **Permanence of the anaesthetist** | | | <0.01 | | <0.001 |
| No | 972 (53.2) | 82 (8.4) | | 245 (28.6) | |
| Yes | 855 (46.8) | 108 (12.6) | | 203 (20.9) | |

*p-value assessing the distribution difference in characteristics between women who preferred CS and women who preferred vaginal delivery.

**p-value assessing the distribution difference in characteristics between women who had a CS and women who had a vaginal birth.

§At least one of the following complications: hypertension and associated complications (n = 127); premature rupture of membranes (n = 223); suspected intrauterine growth retardation (n = 22); type I/II/gestational diabetes (n = 133); heart or kidney disease (n = 9); chronic respiratory conditions (n = 4); HIV (n = 9); cholestasis (n = 4); metrorrhagia (n = 1); condyloma accuminata (n = 4).

**Table 2. Assessment of matching quality by propensity score.**

| | Pseudo-R² | P>chi² | Mean bias | Rubin B indicator | Rubin R indicator |
|---|---|---|---|---|---|
| **Delivery method** | | | | | |
| Unmatched sample | 0.076 | <0.001 | 19.3 | 74.9 | 0.90 |
| Matched sample | 0.003 | 1.00 | 3.5 | 13.9 | 0.99 |

clinicians'practices, or to the organization of the healthcare system [8,11,16,19,54–57]. In Burkina Faso, for example, general practitioners who lacked obstetric skills but worked in maternity wards due to a shortage of specialized staff were more likely to perform unnecessary CS [58]. In Thailand, obstetricians describe vacuum or forceps extraction as requiring "a high level of expertise and skill" that they believe they do not have and consider CS to be the best option to avoid adverse events [17]. In addition, CS is perceived by some doctors as a way to better balance their clinical and academic responsibilities, and personal time, as well as to ensure the baby's safety. In contrast, vaginal delivery is often perceived as more unpredictable in terms of neonatal outcomes [17]. In Viet Nam, healthcare professionals have reported experiencing considerable pressure and being targeted on social media. This results in CS conducted for fear of litigation as CS is seen as a form of protection [16]. This finding is also described in Thailand and Argentina [17,59]. Financial incentives to perform CS rather than vaginal deliveries are regularly mentioned in the literature [8,11,54,59–62]. This is probably the case in Thailand and Viet Nam, where private practice within public hospitals, observed during the research but not yet documented, is widespread. Finally, certain organizational factors may contribute to the use of CS. A previous QUALI-DEC study showed that emergency intrapartum CS was significantly associated with a higher bed occupancy level [48], revealing clinicians' preference for performing CS in order to facilitate the organization of care and, in their opinion, ensure optimal quality of care [11,63,64]. Furthermore, in Argentina, shortage of midwives in some hospitals or their limited role during labor and delivery, while obstetricians manage labor in most public hospitals, even for low-risk pregnancies, are described as factors that may contribute to excessive use of CS [59]. The association observed in our study

**Table 3. Average treatment effect (ATT) of women's preference for CS on the mode of delivery after matching, in the total sample and by country (Quali-Dec).**

| | Proportions before matching | | ATT after matching | | |
|---|---|---|---|---|---|
| | Preference for CS, n (%) | Preference for AVB, n (%) | ATT (CI 95%)§ | p-value | Number of pairs |
| **Total sample (n = 1827)** | | | | | |
| Vaginal birth (n = 1379) | 74 (5.4) | 1305 (94.6) | – | – | |
| CS (n = 448) | 116 (25.9) | 332 (74.1) | +0.32 (0.23; 0.41) | <0.001 | 1709 |
| **By country** | | | | | |
| **Argentina (n = 324)** | | | | | |
| Vaginal birth (n = 257) | 26 (10.1) | 231 (89.9) | – | – | |
| CS (n = 67) | 21 (31.3) | 46 (68.7) | +0.34 (0.17; 0.51) | <0.001 | 323 |
| **Thailand (n = 555)** | | | | | |
| Vaginal birth (n = 404) | 43 (10.6) | 361 (89.4) | – | – | |
| CS (n = 151) | 63 (41.7) | 88 (58.3) | +0.31 (0.19; 0.43) | <0.001 | 555 |
| **Viet Nam (n = 608)** | | | | | |
| Vaginal birth (n = 440) | 2 (0.5) | 438 (99.5) | – | – | |
| CS (n = 168) | 29 (17.3) | 139 (82.7) | +0.58 (0.33; 0.82) | <0.001 | 600 |

§ ATT: Average effect of preference among women who preferred CS (=difference in risk of caesarean delivery between women who preferred CS and those who preferred vaginal birth).

**Table 4. Relative risk* (RR) and fraction of CS associated with women's preference for CS.**

| | RR (CI95%)* | Associated fraction (CI95%) |
|---|---|---|
| **Total sample (n = 1827)** | 2.69 (2.43; 2.95) | 15% (12.9%; 16.9%) |
| **By country** | | |
| Argentina (n = 324) | 2.51 (1.84; 3.17) | 17.8% (10.8%; 23.9%) |
| Thailand (n = 555) | 2.61 (2.17; 3.05) | 23.0% (18.2%; 28.0%) |
| Viet Nam (n = 608) | 3.76 (3.22; 4.29) | 12.0% (10.2%; 14.4%) |

*Relative risk of giving birth by CS for women who had a preference for CS in late pregnancy compared to those who had a preference for vaginal birth, among low-risk women who had a preference in late pregnancy.

between women's preference and mode of delivery is in line with the findings of previous studies in both high-income countries (HICs) [20–23] and in LMICs [20,22]. Our finding may suggest that women who prefer CS are more likely to request it, and some doctors are more inclined to abide by this preference in the absence of medical indications. However, we have no data to indicate whether women's preference resulted in a request for a CS in our study because the practice of CS on demand is not officially accepted in most of the four participating countries. Based on our findings, we assume that in Viet Nam, and to a lesser extent in Argentina and Thailand, women's preference for CS tends to be considered by doctors when deciding on the mode of delivery. In Thailand, for example, from providers' view, CS meets the needs of women and their families while preventing litigation in the event of adverse outcomes in vaginal childbirth [17]. In Argentina, women's preference is supported by a law which, since 2015, guarantees women the right to participate in the decision-making process throughout pregnancy and birth [65].

Although the proportion of CS associated with women's preference for this mode of delivery is low, it varies between countries. The highest proportion of CS associated with women's preference is observed in Thailand and can be explained by the high prevalence of women's preference for CS in that country (19%) compared to other countries. This result had already been mentioned in a previous publication [36], in which we also observed that preference for CS was

more pronounced among nulliparous women and was linked to the fear of pain, especially in Thailand [36]. In this country, pain management is nearly non-existent, and nulliparous women account for more than half (51.6%) of our sample, which could explain the higher prevalence of women's preference for CS. Similarly, there is strong belief in Thailand that the date of birth should be chosen at an auspicious time to ensure the destiny of the child and the family [17].

In Viet Nam, where the proportion of women who prefer CS is low (5%), the strong statistical association between preference and actual mode of birth in Viet Nam could be misleading as doctors were probably the most influential factor in shaping women's preference for CS [36]. The influence of the doctor in the preference for CS is well documented in the literature [13,14,22,53,66]. Several studies in Iran, Brazil and Lebanon, have shown that women who requested a CS were influenced in their decision by their doctor, who reinforced their feeling that a CS was the safest option [13,66–69]. Especially in Viet Nam, this raises questions about the quality of information given by clinicians to their patients. According to the principles of ethical care, healthcare professionals must provide valid information that reflects scientific progress, and is free from personal beliefs or interests that may influence patients' decision [70].

Taking patients' preferences into account is one of the fundamental elements of shared medical decision-making [71,72]. Shared decision-making, which limits inappropriate interventions and reduces the heterogeneity of medical practice, is a key component of the quality of obstetric care. Involving patients in decision-making also improves their satisfaction with care and their knowledge [53,70,71,73,74]. However, it is not just a question of meeting women's expectations, regardless of the risks and benefits of each delivery mode. This shared decision-making process includes the obligation to provide patients with impartial and valid information and to engage in a balanced dialogue that considers the woman's values and perspectives, to reach consensus of the course of action [71,72,75].

Our findings have implications for clinical practice. Low-risk women who reveal a preference for CS during their pregnancy may be interested to use the QUALI-DEC decision analysis tool (DAT) which will help them to make an informed choice [35,76]. This DAT may improve patient-provider communication. But given the relatively small proportion of CS associated with this preference, its potential impact on CS rates could be limited if this DAT is used alone, not in combination with other interventions targeting healthcare providers or health care systems. Assuming the challenge of its widespread use is met, this DAT could help reduce the overall use of CS by encouraging clinicians to base their decisions on the risks and benefits of each mode of delivery and to consider the preferences of most women for vaginal birth. However, our findings also highlight the need to design and implement multidimensional and context-specific strategies that are consistent with WHO recommendations on non-clinical interventions to reduce unnecessary CS rates [28]. These recommendations emphasize the importance of simultaneously addressing as many non-clinical factors as possible that may influence the excessive use of CS. According to the WHO, providing women with evidence-based information, strengthening clinicians' adherence to guidelines, and promoting shared decision-making are effective interventions when adapted to the local context. In addition, and aligned with WHO recommendations, our findings confirm that it is essential to ensure that women receive adequate support during pregnancy and access to appropriate methods of pain relief during labor and delivery. These interventions reduce fear and anxiety associated with pain and childbirth, thereby reducing the preference for CS [77].

To our knowledge, this is the first study to quantify the extent of the influence of women's preference on their mode of birth, measured in a comparable and rigorous manner in four LMICs. The multisite and multicountry design is a strength of this study, as it could allow the results to be generalized to similar contexts, although it should be noted that our non-probabilistic sampling technique limits this generalizability. In addition, we carried out a comprehensive data collection to adjust the analyses on the individual and institutional variables that influence the use of CS. The quality of the survey data was assured based on the WHO data collection and management system for the surveillance of maternal and perinatal health [78]. Finally, the analysis methodology used enabled us to control confounding biases in order to reliably measure the association between women's preference for CS and their mode of delivery.

There are also several limitations to this analysis. The first is that women were questioned in post-partum about their preference in late pregnancy. This approach presents a risk of recall bias, compared with prospective studies in which women are interviewed during pregnancy [24]. In addition, the stated preference may have been influenced by the outcome of their recent childbirth. Women may have been biased toward reporting a preferred mode of delivery consistent with the type of delivery they had just experienced, thereby inflating the observed association. We think that adjusting the measure on actual mode of delivery may control partially this recall bias. The more appropriate measure of the association between women's preferences and their mode of delivery would be a prospective study examining women's preferences during pregnancy. Due to budgetary and research organization constraints in the countries, the option of interviewing women twice (in late pregnancy and then post-partum) was not retained. However, several mechanisms were put in place to minimize the disadvantages of the chosen approach, including the preparation of a survey questionnaire reviewed by many experts, tested in each country and used by specially trained interviewers. In addition, we were not able to analyze according to the type of CS (during or before labour), due to a limited number of women. This stratified analysis would have allowed us to test the hypothesis that women's preference for a CS is more likely to lead to a planned CS than an emergency CS during labour. Matched propensity score analyses enabled control for confounding bias. This method has been reported to provide a more accurate estimate of the association between an exposure factor and an outcome and more reliable statistical inference than multivariate logistic regression analysis [38]. However, unmatched women, who tend to have 'extreme' propensity scores (outside the common support), are excluded from the analysis, resulting in a loss of power. In addition, matched women may have similar propensity scores but very different characteristics, as the matching is not based on the variables themselves but on a score combining all these variables. Nor can we state that all the variables explaining both the exposure (women's preference) and the outcome (caesarean delivery) were included in the calculation of the propensity score. Indeed, some likely confounding factors could not be measured, in particular the attitudes and clinical practices of providers and institutional norms are not captured in the data. Finally, we calculated the proportion of CS associated with preference for this mode of birth. However, we cannot definitely establish a causal relationship between women's preferences and their mode of birth, as the data are cross-sectional and do not account for the temporality of events. For this reason, we have not used the term "attributable fraction" but rather "associated fraction". Therefore, the findings on associated fractions provide an indication of the contribution of women's preference to the use of CS and should be interpreted with caution.

## Conclusion

Although women's preference plays a role in the use of CS in the QUALI-DEC participating hospitals, it likely accounts for only a small proportion of CS performed. A more nuanced understanding of this influence is crucial for designing targeted interventions and shaping their focus. These findings reinforce the need for multidimensional interventions that engage all relevant stakeholders. The results suggest the need to provide women with appropriate, evidence-based information, to improve clinicians' adherence to guidelines, and to strengthen shared decision-making, in order to reduce the overuse of CS. It is also necessary to address other determinants, in particular the systemic factors that influence the use of CS. Finally, it is necessary to improve women's experience of childbirth so that they can feel confident in choosing and envisaging vaginal birth.

## Supporting information

**S1 Checklist. Inclusivity in global research.**
(DOCX)

**S2 Table. Institutional variables of the hospitals where the study women delivered (QUALI-DEC).**
(DOCX)

**S3 Fig. Directed acyclic graph representing the link between women's preference and the final mode of birth.**
(TIFF)

## Acknowledgments

We thank all women who participated to the survey and healthcare providers who facilitated the data collection in participating hospitals. We also thank Andrainolo Ravalihasy, a statistical engineer at CEPED unit who advised the first author (C.E.) on statistical analyses, and all the members of the QUALI-DEC consortium group. The composition of this group is as follows: Karolinska Institutet (Stockholm, Sweden): Claudia Hanson, Helle Molsted-Alvesson, Kristi Sidney Annerstedt, Amanda Cleeve; University College Dublin, National University of Ireland (Dublin, Ireland): Michael Robson; World Health Organization (Geneva, Switzerland): Ana Pilar Betrán, Newton Opiyo, Meghan Bohren; Centro Rosario de Estudios Perinatales Asociacion (Rosario, Argentina): Guillermo Carroli; Liana Campodonico; Celina Gialdini; Berenise Carroli; Gabriela Garcia Camacho; Daniel Giordano; Hugo Gamerro; CEDES (Buenos Aires, Argentina): Mariana Romero; Khon Kaen University (Khon Kaen, Thailand): Pisake Lumbiganon, Dittakarn Boriboonhirunsarn, Nampet Jampathong, Kiattisak Kongwattanakul, Ameporn Ratinthorn, Olarik Musigavong; Fundacio Blanquerna (Barcelona, Spain): Ramon Escuriet, Olga Canet; Centre national de recherche scientifique et technologique—Institut de Recherche en sciences de la santé (Ouagadougou, Burkina Faso): Charles Kabore, Yaya Bocoum Fadima, Simon Tiendrébéogo, Zerbo Roger; Pham Ngoc Thach University of Medicine (Ho Chi Minh, Viet Nam): Mac Quoc Nhu Hung, Thao Truong, Tran Minh Thien Ngo, Bui Duc Toan, Huynh Nguyen Khanh Trang, Hoang Thi Diem Tuyet; Research Institute for Sustainable Development (Paris, France): Alexandre Dumont, Laurence Lombard, Myriam de Loenzien, Marion Ravit, Delia Visan. The lead author for the QUALI-DEC group is Alexandre Dumont (contact: alexandre.dumont@ird.fr)

## Author contributions

**Conceptualization:** Camille Etcheverry, Ana Pilar Betrán, Marion Ravit, Charles Kaboré, Pisake Lumbiganon, Guillermo Carroli, Quoc Nhu Hung Mac, Celina Gialdini, Alexandre Dumont.

**Data curation:** Camille Etcheverry, Marion Ravit.

**Formal analysis:** Camille Etcheverry.

**Funding acquisition:** Alexandre Dumont.

**Investigation:** Charles Kaboré, Pisake Lumbiganon, Guillermo Carroli, Quoc Nhu Hung Mac, Celina Gialdini, Alexandre Dumont.

**Supervision:** Ana Pilar Betrán, Alexandre Dumont.

**Validation:** Ana Pilar Betrán, Alexandre Dumont.

**Writing – original draft:** Camille Etcheverry.

**Writing – review & editing:** Camille Etcheverry, Ana Pilar Betrán, Marion Ravit, Charles Kaboré, Pisake Lumbiganon, Guillermo Carroli, Quoc Nhu Hung Mac, Celina Gialdini, Alexandre Dumont.

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
