## [Decision Letter · Decision Letter 0]

22 Jun 2025

Dear Dr. Etcheverry,

Thank you for submitting your manuscript to PLOS ONE. After careful consideration, we feel that it has merit but does not fully meet PLOS ONE’s publication criteria as it currently stands. Therefore, we invite you to submit a revised version of the manuscript that addresses the points raised during the review process.

We look forward to receiving your revised manuscript.

Kind regards,

Mengstu  Asaye, PhD 

Academic Editor

PLOS ONE

Journal Requirements:

[The QUALI-DEC project is co-funded by the European Union’s Horizon 2020 research and innovation program under grant agreement No. 847567 and by the UNDP-UNFPA-UNICEF-WHO-World Bank Special Programme of Research, Development and Research Training in Human Reproduction (HRP), a cosponsored program executed by the World Health Organization (WHO) in the Department of Sexual and Reproductive Health and Research (SRH). The contents of this article are solely the responsibility of the authors and do not reflect the views of the EU, UNDP, UNFPA, UNICEF, WHO, or the World Bank or their respective institutions. The first author (C.E.) received salary support from the Ecole Doctorale Pierre Louis de Santé Publique (Sorbonne Université Université Paris Cité) as part of a thesis funding.].

6. One of the noted authors is a group or consortium [QUALI-DEC research group]. In addition to naming the author group, please list the individual authors and affiliations within this group in the acknowledgments section of your manuscript. Please also indicate clearly a lead author for this group along with a contact email address.

7. Your ethics statement should only appear in the Methods section of your manuscript. If your ethics statement is written in any section besides the Methods, please move it to the Methods section and delete it from any other section. Please ensure that your ethics statement is included in your manuscript, as the ethics statement entered into the online submission form will not be published alongside your manuscript.

Reviewers' comments:

Reviewer's Responses to Questions

**Comments to the Author**

1. Is the manuscript technically sound, and do the data support the conclusions?

Reviewer #1: Yes

Reviewer #2: Yes

2. Has the statistical analysis been performed appropriately and rigorously?

Reviewer #1: Yes

Reviewer #2: Yes

3. Have the authors made all data underlying the findings in their manuscript fully available?

Reviewer #1: Yes

Reviewer #2: Yes

4. Is the manuscript presented in an intelligible fashion and written in standard English?

Reviewer #1: Yes

Reviewer #2: Yes

Reviewer #1: The manuscript presents a well-conducted multi-country study addressing the critical issue of rising cesarean section (CS) rates in low- and middle-income countries (LMICs). The use of propensity score matching (PSM) to control for confounders strengthens the analysis, and the findings—that women’s preference accounts for only 15% of CS—provide valuable evidence challenging the common perception that maternal demand is a primary driver of CS overuse. The study’s multi-country design enhances its generalizability to similar LMIC contexts. However, the following revisions should be addressed to improve clarity and robustness:

1. Clarify Causal Inference:

o While PSM reduces confounding, the manuscript overstates causal claims (e.g., "the fraction of CS attributable to women’s preference"). Emphasize that the design is observational and that unmeasured confounders (e.g., clinician influence, financial incentives) may bias results. Consider using terms like "association" rather than "attributable."

2. Address Recall Bias:

o Women’s preferences were assessed postpartum, which risks recall bias and reverse causality (preference influenced by delivery outcome). Expand the discussion of how this might inflate the observed association and suggest future studies prospectively measure preferences during pregnancy.

3. Contextualize Country-Specific Findings:

o The high attributable fraction in Thailand (23%) and stark contrast in Vietnam (12%) warrant deeper exploration. Discuss potential cultural, institutional, or policy differences (e.g., Vietnam’s clinician influence on preferences) that might explain these variations.

4. Strengthen the Discussion of Clinician/Systemic Factors:

o While the conclusion mentions systemic factors, the manuscript underanalyzes how clinician preferences or hospital policies (e.g., financial incentives, fear of litigation) interact with women’s preferences. Cite qualitative findings from the QUALI-DEC project or existing literature to contextualize these dynamics.

5. Data Availability:

o The statement that data will be available by late 2024 is acceptable, but clarify how readers can access the data post-publication (e.g., Zenodo DOI, project website).

6. Minor Revisions:

o Tables/Figures: Ensure consistency in terminology (e.g., "Viet Nam" vs. "Vietnam"). Simplify Table 1 for readability (e.g., merge redundant categories).

o Abstract: Specify the timeframe of data collection (2020–2022) to contextualize the study period.

o Introduction: Briefly define "low-risk" women using Robson classification criteria to aid non-specialist readers.

Overall, the manuscript makes a meaningful contribution to understanding CS overuse in LMICs and merits publication after addressing the above revisions. Its findings underscore the need for holistic interventions targeting clinicians, health systems, and patient education, which aligns with global maternal health priorities.

Reviewer #2: Review comments

First I would like to say thank you the editors for invited me to review this article entitled “Contribution of women's preference to the overuse of caesarean section: a propensity score matching analysis based on a multi-country cross-sectional survey, as part of the QUALI-DEC project.” With Manuscript Number: PONE-D-25-15077

1. Generally it is well written paper and targeted its objectives.

2. Used advanced English language

3. However I have some question section by section

Abstract

1. Your title and your objectives in abstract section are not similar? Why? Line#33

2. You wrote CS was high due to maternal request, if it was high but your argument was lack of evidence it was paradox; why you want to conduct this title? Line #32

3. The sample size was not clear because it seems goes to numbers of stages; in addition you took 71 participants in each hospital. If so 71*32=2272 but your sample size was 1827. Why? It needs deep explanation.

Introduction

4. The argument you stated is convincing. Why you want to conduct? How we know you were free from the projects influence that may have impact on the result of this paper?

Methods and materials

5. How you selected the study areas from other LMICs? It did not show the probability sampling technique. If so your generalization of the results of this paper may in questioned?

6. Your participant in each hospital was 71 for minimum of two-week. Then you stated that if you reached the total sample (71) before two-weeks you continued your data collection till two-weeks. Why? Line# 182.

7. This sentence is clear for me “For high-activity 186 hospitals, a randomization factor was applied each day to all women who had given birth the 187 previous day to obtain a random sample of 10 women, assuming that between 4 and 5 women 188 would refuse to participate or would not be eligible.” Line# 186-188.

Result: No need of revision.

Discussion: I think it was your limitation “The multi-site and multi-country design is a strong point of this study, allowing the 420 results to be generalized to similar contexts.” Because your sampling technique was not purely probable type. You used nonprobability (purposive) type.

**Do you want your identity to be public for this peer review?** For information about this choice, including consent withdrawal, please see our Privacy Policy

Reviewer #1: **Yes: ** Afework Tadele

Reviewer #2: No

---

## [Author Response · Author response to Decision Letter 1]

20 Jul 2025

Responses to editor's comments:

1. PLOS ONE style requirements: We have made corrections to bring the manuscript into compliance with PLOS ONE style requirements, including those relating to file naming.

2. PLOS Global Research Inclusivity Questionnaire: The completed PLOS Global Research Inclusivity Questionnaire has been provided as supplementary information with the revised manuscript (S1 Checklist).

3. Funding statement: We have declared all funding or sources of support in the funding statement and have also included the statement “No additional external funding was received for this study” in our updated funding statement.

4. “Funding information” and “Financial disclosure” sections: We have made the requested corrections to ensure that these two sections match.

5. Data availability: We have clarified our statement on data availability as requested. The data analyzed in this study are part of the data generated by the Quali-Dec project and are still being analyzed by the research consortium. These data cannot therefore be shared at this time, but will be available at the end of the project in late 2025. We have provided a link to the community “QUALI-DEC - Appropriate use of Caesarean section through QUALIty DECision-making by women and providers (847567)” on Zenodo, where the data will be published. The database anonymization process is ongoing and is being managed by the project data manager. All data relating to the Quali-Dec survey will be available at the end of the project (postponed to the end of 2025), as set out in the research protocol approved by the ethics committees and the European Union. The data sharing plan has already been defined in accordance with best practices and will be implemented as planned. However, the data underlying the results presented in the study will be available upon reasonable request to the corresponding author, if the research consortium agrees. Zenodo is a general open-access repository developed as part of the European OpenAIRE program and managed by the European Organization for Nuclear Research (CERN). Zenodo will allow the deposit of datasets, reports, and any other digital artifacts related to the research. For each deposit, a permanent DOI will be created to facilitate citation of the stored items. The metadata for each record will be indexed and searchable directly in the Zenodo search engine immediately after publication.

6. One of the authors mentioned is a group or consortium [QUALI-DEC research group]: We have listed the individual authors and their affiliations within this group in the “Acknowledgments” section and provided the email address of the lead author.

7. Ethical statement: We have added the ethical statement in the “Methods” section, as requested.

8. References: We have checked that the references are correct.

Responses to reviewers' comments:

We have provided a document called “responses to reviewers” where we respond point by point to each reviewer's questions. We have clarified certain information as requested and made changes to the manuscript as recommended by the reviewers where appropriate. These changes are visible in the revised manuscript with track changes.

Responses to Reviewer #1:

1. Clarify Causal Inference:

We agree that the observational design does not allow us to claim a causal link. We have therefore used the term “associated” (e.g., “fraction of CS associated with women's preference”) instead of “attributable,” as you recommended.

This limitation is emphasized in the following sentences in the discussion section:

“However, we cannot definitely establish a causal relationship between women's preferences and their mode of birth, as the data are cross-sectional and do not account for the temporality of events. For this reason, we have not used the term “attributable fraction” but rather “associated fraction”. Therefore, the findings on associated fractions provide an indication of the contribution of women's preference to the use of CS and should be interpreted with caution.”

Page 21-22, Lines 507-512 of the revised manuscript (without track changes)

2. Address Recall Bias:

We have expanded the discussion as recommended, with the following sentences:

“Women may have been biased toward reporting a preferred mode of delivery consistent with the type of delivery they had just experienced, thereby inflating the observed association. We think that adjusting the measure on actual mode of delivery may control partially this recall bias. The more appropriate measure of the association between women's preferences and their mode of delivery would be a prospective study examining women’s preferences during pregnancy”

Page 21, Lines 483-488 of the revised manuscript (without track changes)

3. Contextualize Country-Specific Findings:

Indeed, the proportion of CS associated with women's preferences varies across countries. The higher proportion of CS associated with women's preferences observed in Thailand is explained by the high prevalence of women's preference for CS in that country (19%) compared to Vietnam, where it is only 5%. This finding had already been mentioned in a previous Quali-DEC publication, in which we also observed that preference for CS was more pronounced among nulliparous women and was linked to the fear of pain, especially in Thailand. In this country, pain management is nearly non-existent, and nulliparous women account for more than half (51.6%) of our sample, which could explain the higher prevalence of women's preference for CS. Similarly, there is a strong belief in Thailand that the date of birth should be chosen at an auspicious time to ensure the destiny of the child and the family.

As the reviewer suggested, we have expanded the discussion section to incorporate these explanations. Please, see pages 18-19, Lines 425-435 of the revised manuscript (without track changes). We have also added references to the previous QUALI-DEC analysis.

4. Strengthen the Discussion of Clinician/Systemic Factors:

We drew on previous qualitative studies to explain how clinician-related or systemic factors may explain the results. We revised the following section in the discussion:

“In Burkina Faso, for example, general practitioners who lacked obstetric skills but worked in maternity wards due to a shortage of specialized staff were more likely to perform unnecessary CS. In Thailand, obstetricians describe vacuum or forceps extraction as requiring “a high level of expertise and skill” that they believe they do not have and consider CS to be the best option to avoid adverse events. In addition, CS is perceived by some doctors as a way to better manage their clinical and academic work and personal time, but also to ensure the baby’s safety, while vaginal delivery is perceived as more unpredictable in terms of neonatal outcomes. In Vietnam, healthcare professionals have reported experiencing considerable pressure, which explains why CS is often decided for fear of litigation or being targeted on social media, as CS is seen as a form of protection. This finding is also described in Thailand and Argentina. Financial incentives to perform CS rather than vaginal deliveries are regularly mentioned in the literature. This is probably the case in Thailand and Vietnam, where private practice within public hospitals, observed during the research but not yet documented, is widespread. Finally, certain organizational factors may contribute to the use of CS. A previous QUALI-DEC study showed that emergency intrapartum CS was significantly associated with a higher bed occupancy level, revealing clinicians' preference for performing CS in order to facilitate the organization of care and, in their opinion, ensure optimal quality of care. Furthermore, in Argentina, shortage of midwives in some hospitals or their limited role during labor and delivery, while obstetricians manage labor in most public hospitals, even for low-risk pregnancies, are described as factors that may contribute to excessive use of CS”

Page 17-18, Lines 390-411 of the revised manuscript (without track changes)

5. Data Availability:

The publication of Quali-Dec data is postponed to the end of 2025, as the data are still being analyzed (before-after evaluation). However, the data sharing plan is a requirement of the research protocol and will be applied as soon as possible.

We have added details on how readers can access the data. We have provided the web link that leads directly to the QUALI-DEC community on Zenodo, where the data will be published.

6. Minor Revisions:

o We have ensured consistency in terminology when referring to Viet Nam. We have aligned ourselves with the terminology policy of the WHO and the United Nations, which use the two-word spelling “Viet Nam” in English. This terminology has been chosen to respect the official preference of the Vietnamese government. The two-word form is considered more respectful of the country's linguistic and cultural conventions.

o Table 1 has been simplified as recommended (pages 13 and 14)

o Abstract: We have added the data collection period (2020-2022) in the abstract. Page 2, Line 37

o Definition of low-risk women: “Women belonging to groups 1 to 4 of the Robson classification are the target population for Quali-Dec [29]. They are at lower risk of CS, as compared to groups 5, or groups 6 to 10.” Page 4, Lines 108-109

Responses to Reviewer #2

1. Abstract:

The objective has been harmonized between the abstract and the introduction.

2. Abstract:

The increasing CS rates are often perceived to be associated with an increase in maternal demand for CS, as reported by many clinicians interviewed in studies available in scientific literature. However, this perception may not reflect reality, and current evidence supporting this claim is limited. Few hospital-based studies have investigated the link between maternal demand and CS rates, but these results are biased because maternal demand is usually poorly reported. For this reason, we thought that our study was important to fill the knowledge gap about the contribution of women’s preference and demand for CS in the increasing CS rates by measuring the real proportion of CS associated with women’s preference.

3. Abstract:

You are correct, the total sample was larger than the sample analyzed in this study. As shown in the flowchart, a total of 3,127 women were included in the survey. However, to reduce bias, we restricted our analysis to a sample of low-risk women defined as women with no history of CS, singleton pregnancy, cephalic presentation, and term delivery – 37 or more weeks of pregnancy), i.e. 1,827 women (also indicated in the flowchart).

4. Introduction:

Let me rephrase our argument. The commonly shared idea that rising CS rates are associated with increased maternal demand for CS was not supported by strong evidence. Few hospital-based studies have investigated the link between maternal demand and CS rates, but these results are biased because maternal demand is usually poorly reported. In participating countries of Quali-Dec project, CS on maternal request is not allowed. Measuring women's preference for CS rather than maternal demand is therefore more accurate in this context. By examining the proportion of CS associated with women’s preference, we thought we could obtain a good approximation of the proportion of CS performed at maternal request and confirm or refute the hypothesis that women's preference or demand for CS contributes to higher CS rates.

All researchers analyzing the data have no conflict of interest and are completely independent from both the funders and the management of the participating hospitals. This study was conducted as a secondary analysis following questions raised by the co-authors during the analysis of collected data. This study was neither commissioned nor influenced by external institutions or organizations. It stems from the researchers' desire to understand the context in which the QUALI-DEC research is being conducted for scientific purposes.

5. Methods and materials:

The QUALI-DEC study is an implementation research. The participating countries were chosen because of the concern of local scientific and medical authorities about the significant increase in CS rates in many of their hospitals and its consequences. In addition, we found a clear determination among policymakers in these countries to revert this trend using evidence-based interventions. This project is also the result of a long collaboration between the researchers involved in this subject, who currently make up the QUALI-DEC consortium. The choice of these countries was therefore operational. A detailed description of the rational, methods and Theory of Change has been published in two scientific manuscripts reporting the protocols of the study (effectiveness and process evaluation protocols):

Cleeve et al., 2023, Global Health Action: https://www.tandfonline.com/doi/full/10.1080/16549716.2023.2290636

Dumont et al., 2020, Implementation Science: https://implementationscience.biomedcentral.com/articles/10.1186/s13012-020-01029-4

Indeed, the sampling technique was not probabilistic, which limits the generalizability of the results. We therefore qualified this point in the discussion:

“The multisite and multicountry design is a strength of this study, as it could allow the results to be generalized to similar contexts, although it should be noted that our non-probabilistic sampling technique limits this generalizability.”

Page 20, Lines 470-472 of the revised manuscript (without track changes)

6. Methods and materials:

That's right. Methodologically, it was necessary to have the same two-week collection period at each site to ensure a consistent procedure everywhere. When the sample size was not reached within two weeks, we prolonged the data collection period. This was necessary in 8 hospitals.

This approach has been used before in surveys: Bull World Health Organ. 2007 Nov 26;86(2):126–131. doi: 10.2471/BLT.06.039842

7. Methods and materials:

We have changed this sentence for better clarity:

“In hospitals with more than 10 deliveries per day, we randomly selected 10 women each day among all women who had given birth the previous day. Assuming that 4 to 5 women would refuse to participate or would not be eligible from this random sample, 5 to 6 women would be included in the survey each day, allowing the required number of subjects to be reached.”

Page 8, Lines 204-209

8. Discussion:

We have nuanced this point and mentioned the limitation of the sampling technique as explained in Page 20, Lines 470-472 of the revised manuscript (without track changes)

---

## [Decision Letter · Decision Letter 1]

2 Oct 2025

Dear Dr. Etcheverry,

Thank you for submitting your manuscript to PLOS ONE. After careful consideration, we feel that it has merit but does not fully meet PLOS ONE’s publication criteria as it currently stands. Therefore, we invite you to submit a revised version of the manuscript that addresses the points raised during the review process.

The manuscript has been evaluated by three reviewers and their comments are available below. 

The reviewers have raised a number of concerns that need attention. They request some updates to the introduction, as well as clarification on the validity of the tools used.

Could you please revise the manuscript to carefully address the concerns raised?

We look forward to receiving your revised manuscript.

Kind regards,

Jen Edwards

Staff Editor

PLOS ONE

Journal Requirements:

Reviewers' comments:

Reviewer's Responses to Questions

**Comments to the Author**

Reviewer #1: All comments have been addressed

Reviewer #2: All comments have been addressed

Reviewer #3: (No Response)

2. Is the manuscript technically sound, and do the data support the conclusions?

Reviewer #1: Yes

Reviewer #2: Yes

Reviewer #3: Yes

3. Has the statistical analysis been performed appropriately and rigorously?

Reviewer #1: Yes

Reviewer #2: Yes

Reviewer #3: Yes

4. Have the authors made all data underlying the findings in their manuscript fully available?

Reviewer #1: Yes

Reviewer #2: Yes

Reviewer #3: Yes

5. Is the manuscript presented in an intelligible fashion and written in standard English?

Reviewer #1: Yes

Reviewer #2: Yes

Reviewer #3: Yes

Reviewer #1: My first comments were thoroughly addressed and carefully considered, leaving me genuinely satisfied with the response.

Reviewer #2: All my comments were properly addressed. Therefore, I have no additional comments and questions to be addressed again.

Reviewer #3: in your abstract section, abbreviation is not recommended, so omit it

in introduction section, if so what interventions were done to decrease unnecessary C/S worldwide, in low and middle income countries specifically in your study area?

line 60 how increasing rate of C/S affect countries? in which context? line 87 odds of previous studies is not necessary,

why women's who gave stillbirth or neonatal death before discharge excluded form your study?

during baseline survey data collection period was two weeks in each hospital with the required number of (n=71) for two weeks in each hospital, during this time if the required number is reached before two weeks duration, data collection was continued until the end of predefined period.

if the tools were prepared by reviewing previous literatures how did you check the validity.

**Do you want your identity to be public for this peer review?** For information about this choice, including consent withdrawal, please see our Privacy Policy

Reviewer #1: **Yes: ** Afework Tadele

Reviewer #2: No

Reviewer #3: No

---

## [Author Response · Author response to Decision Letter 2]

22 Oct 2025

Thank you for your peer review. We have improved this article in accordance with your recommendations. In particular, we have added some updates to the introduction and clarified the validity of the tools used. All changes and responses to comments are presented in the “responses to reviewers” file.

---

## [Decision Letter · Decision Letter 2]

10 Nov 2025

Dear Dr. Etcheverry,

Thank you for submitting your manuscript to PLOS ONE. After careful consideration, we feel that it has merit but does not fully meet PLOS ONE’s publication criteria as it currently stands. Therefore, we invite you to submit a revised version of the manuscript that addresses the points raised during the review process.

We look forward to receiving your revised manuscript.

Kind regards,

Miquel Vall-llosera Camps

Senior Staff Editor

PLOS ONE

Journal Requirements:

Reviewers' comments:

Reviewer's Responses to Questions

**Comments to the Author**

Reviewer #1: All comments have been addressed

Reviewer #3: (No Response)

2. Is the manuscript technically sound, and do the data support the conclusions?

Reviewer #1: Yes

Reviewer #3: (No Response)

3. Has the statistical analysis been performed appropriately and rigorously?

Reviewer #1: Yes

Reviewer #3: (No Response)

4. Have the authors made all data underlying the findings in their manuscript fully available?

Reviewer #1: Yes

Reviewer #3: (No Response)

5. Is the manuscript presented in an intelligible fashion and written in standard English?

Reviewer #1: Yes

Reviewer #3: (No Response)

Reviewer #1: No major modifications are necessary. However, to further strengthen clarity and precision before final submission, consider:

Abstract (lines 29–53):

Suggest simplifying the phrasing in the final sentence to make the policy implication more direct (e.g., “...highlighting the need for multidimensional, context-specific strategies to reduce unnecessary caesarean sections.”)

Statistical section (lines ~244–317):

Add a brief explanation of why the nearest-neighbour matching with replacement was chosen over caliper or kernel methods (for transparency and reproducibility).

Discussion:

The interpretation is comprehensive, but the discussion could briefly note the potential for unmeasured confounding, particularly provider attitudes or institutional norms not captured in the data.

A short paragraph linking the study’s findings to WHO’s recommendations on CS reduction could strengthen its applied relevance.

Formatting:

Ensure consistent spacing and punctuation (minor spacing issues appear intermittently in the Results tables and references).

Verify all references are formatted per PLOS ONE style (check capitalization, italics, and DOI availability).

Reviewer #3: (No Response)

**Do you want your identity to be public for this peer review?** For information about this choice, including consent withdrawal, please see our Privacy Policy

Reviewer #1: **Yes: ** Afework Tadele

Reviewer #3: No

---

## [Author Response · Author response to Decision Letter 3]

28 Nov 2025

Thank you for your review. In accordance with your recommendations, we have added a brief explanation of the choice of matching method ( Methods section), potential unmeasured confounders (Discussion - Limitations section), and the link between the study's conclusions and WHO recommendations on reducing cesarean sections (Discussion section). Consistency in spacing and punctuation has been respected, as well as the format of references.

In addition, all data used in this study have been published since the last revision, and we have updated our data availability statement.

---

## [Editor Report · Decision Letter 3]

1 Dec 2025

Contribution of women's preference to the overuse of caesarean section: a propensity score matching analysis based on a multi-country cross-sectional survey, as part of the QUALI-DEC project

PONE-D-25-15077R3

Dear Dr. Etcheverry,

We’re pleased to inform you that your manuscript has been judged scientifically suitable for publication and will be formally accepted for publication once it meets all outstanding technical requirements.

Kind regards,

Miquel Vall-llosera Camps

Senior Staff Editor

PLOS One
---

## [Editor Report · Acceptance letter]

PONE-D-25-15077R3

PLOS One

Dear Dr. Etcheverry,

I'm pleased to inform you that your manuscript has been deemed suitable for publication in PLOS One. Congratulations! Your manuscript is now being handed over to our production team.

Kind regards,

on behalf of

Dr Jen Edwards

Staff Editor

PLOS One